# Nutritional Quality, Chemical, and Functional Characteristics of Hemp *(Cannabis sativa ssp. sativa*) Protein Isolate

**DOI:** 10.3390/plants11212825

**Published:** 2022-10-24

**Authors:** Sobhy Ahmed El-Sohaimy, Natalia Vladimirovna Androsova, Abduvali Djabarovich Toshev, Hesham Ali El Enshasy

**Affiliations:** 1Department of Technology and Organization of Public Catering, Institute of Sport Tourism and Service, South Ural State University, 4544080 Chelyabinsk, Russia; 2Department of Food Technology, Arid Lands Cultivation Research Institute, City of Scientific Research and Technological Applications, Alexandria 21934, Egypt; 3Institute of Bioproduct Development (IBD), Universiti Teknologi Malaysia (UTM), Skudai 81310, Malaysia; 4Faculty of Chemical and Energy Engineering, Universiti Teknologi Malaysia (UTM), Skudai 81310, Malaysia; 5City of Scientific Research and Technology Applications (SRTA), New Burg Al Arab, Alexandria 21934, Egypt

**Keywords:** *Cannabis sativa ssp. sativa*, protein isolate, isolation conditions, amino acid composition, chemical properties, functional properties

## Abstract

(1) **Background**: Hemp seeds are a source of plant-based protein, making them an appropriate supplement to a plant-based diet. The current work was focused on the preparation of the protein isolate from the hemp seeds with eco-friendly and cheap technology. Moreover, it evaluated the physicochemical and functional properties of hemp protein isolate for its potential application in food manufacturing. (2) **Methods:** The protein content of hemp seeds has been isolated through two main steps: (1) extraction of the protein content of an alkaline pH (10–12); (2) precipitation of the extracted protein on an acidic pH as an isoelectric point (pH = 4.5). (3) **Results:** The edastin protein is the most predominant protein in the protein profile with a molecular weight of 58.1 KDa beside albumin with a molecular weight of 31.5 KDa. The FTIR spectrum detected the absorption peaks of the amide I at 1750 and 1600 cm^−1^, which pointed to C=O stretching while N-H stretching at 1650–1580 cm^−1^. The peak at 3250 is found to be related to N-H stretching of the aliphatic primary amine (3400–3300 cm^−1^) and the N-H stretching for the secondary (II) amine appeared at 3350–3310 cm^−1^. The Hemp protein isolate (HPI) showed a high content of arginine (15.52 g/100 g), phenylalanine + tyrosine (9.63 g/100 g), methionine + cysteine (5.49 g/100 g), leucine + isoleucine (5.21 g/100 g), and valine (4.53 g/100 g). It contains a moderate level of threonine (3.29 g/100 g) and lysine (2.50 g/100 g) with tryptophan as the limiting amino acid (0.22 g/100 g). The HPI showed an appropriate water-and-oil holding capacity (4.5 ± 2.95 and 2.33 ± 1.88 mL/g, respectively). The foaming capacity of the HPI was increased with increasing the pH values to reach the maximum value at pH 11 (67.23 ± 3.20%). The highest emulsion ability index of the HPI was noted at pH 9 (91.3 ± 2.57 m^2^/g) with low stability (19.15 ± 2.03). (4) **Conclusions:** A strong positive correlation (r = 0.623) was shown between protein concentration and solubility. The current easy-to-use, cheap, and eco-friendly technology provides the industrial sector with a cheap protein isolate for manufacturing protein-rich diet and beverages. The HPI showed a good nutritional quality and functional properties that might be helpful in utilizing it in different food products such as beverages and bakery products.

## 1. Introduction

Hemp *(Cannabis sativa ssp. sativs*) is one of the oldest plants cultivated in Asian countries. Hemp has a great economic importance as it was used to produce textiles, paper, clothing, and household items. Hemp has been used for medical and food purposes as well [1]. In Russia, hemp has been cultivated along with other crops such as flax [2]. However, cultivated areas declined sharply after the UN adopted the single convention on narcotic drugs in 1961. After that, breeders were thrown into the creation of *cannabis* varieties with a low content of tetrahydrocannabinol (THC) (non-narcotic hemp). The varieties of non-narcotic hemp (USO 14, USO 11, USO 31) were cultivated for the first time in Ukraine. To date, 28 varieties of the industrial hemp with low THC content are allowed for cultivation [2]. The seeds of *Cannabis sativa* are round, dark red or brown in color, 3 to 5 mm in diameter. Each grain is covered with a thin pericarp, and dicotyledons [3]. Hemp seeds contain about 18–23% protein, 25–30% oil, 30–40% fiber and 6–7% moisture [1,4]. Several recent studies declared that about 181 proteins were identified in hemp seeds with the main storage proteins edestin and globulin in a concentration of 67% to 75% and globular albumin that ranged from 25 to 37% [5]. The edestin protein of hemp is superior to albumin and contains a high sulfur amino acid content (methionine and cystine) [6]. Hemp protein isolate (HPI) is a source of all nine essential amino acids with a high content of arginine and glutamic acid while containing a moderate amount of sulfur-containing amino acids [7]. On the other hand, the hemp seeds have about 30–35% of lipid content with 90% of unsaturated fatty acids [8]. Hemp oil is characterized as a rich source of essential fatty acids, especially linoleic acid, α-linolenic acid and oleic acid [9,10]. The high level of polyunsaturated fatty acids can contribute to reduce the risk of cardiovascular disease, cancer, rheumatoid arthritis, hypertension, inflammatory, and autoimmune diseases [1,11]. Furthermore, hemp fiber is located in the shell of hemp seeds which contains both soluble and insoluble dietary fiber in a ratio of 20:80 [10,12]. Unlike its soluble counterpart, insoluble fiber does not form a gel with water; thus, its digestion in the upper gastrointestinal tract is very limited. The insoluble fiber of hemp seeds contains cellulose (46%), lignin (31%) and hemicellulose (22%) [8,13]. The phenolic content of hemp seeds is dependent on the variety and the part of the seeds. The highest concentration of phenolic compounds was noted in the shells while the lowest concentration was in the core [14]. The lignan amides and hydroxy cinnamic acid are the predominant phenolic compounds that are showing a high antioxidant capacity in the hemp [14]. Hemp seeds contain a low level of tetrahydrocannabinol (THC) (less than 1%) [4]. On the other hand, hemp seeds contain several anti-nutritional factors that interfere with nutrition, such as trypsin inhibitors and phytic acid [15]. Hemp seed flour is utilized by several researchers around the world in food applications. Shivani and Prabhasankar [16] showed that the addition of hemp flour led to an increase in the protein, polyunsaturated fatty acids, and fiber content, while the structural and mechanical properties of the dough worsened, and its moisture-binding capacity was decreased. Hemp flour has been used as a partial replacement of wheat flour at levels of 10% and 20% for developing gluten-free bread. The incorporation of hemp flour with wheat flour resulted in increasing the nutritional value of the bread, with slight changes in the structural and mechanical properties. The increase in the volume and the slowdown in the processes of the staling of the bread led to the extension in the shelf life of the bread [17]. Replacing the wheat flour with hemp seed flour in bakery products resulted in an increase in the content of protein and polyphenolic compounds, while it led to a decrease in the volume of bread and the rate of staling [18]. Nowadays, alternative protein sources receive serious attention in recent research all over the world. Previous research targeting the effectiveness of the different conditions of protein recovery, quantity, and quality. El-Sohaimy et al. [19,20] studied the effect of pH values ranged (6–12) and mineral salts (NaCl, Na_2_ SO_4_ and MgCl) on the extraction level of the protein from chickpea, lupin, and lentil seeds, followed by the evaluation of different precipitating agents (Pi, Ammonium sulfate, ethanol, methanol and trichloro acetic acids) and concluded that the maximum solubilization and extractability of plant proteins was on an alkaline pH value [10,11,12] and the precipitation of most protein content was at the isoelectric point (pi) (pH 4.6–4.5). These conditions are counted as the optimal protein isolation condition that might be used for large scale production of plant protein isolate for food processing and nutrition purposes. El-Sohaimy et al. [19] showed that the optimum conditions for preparing the protein isolate from quinoa seeds was obtained by protein solubility at the alkaline pH (10) followed by precipitation at an acidic pH value (4.5) as an isoelectric point (pI). El-Sohaimy et al. [20] prepared the protein isolate from chickpea for the fortification of muffins with the solubilization at pH 11 and the solubilized protein was precipitated at pH 4.5 as an isoelectric point (Pi). Due to the quality of hemp protein and technology development and based on our previous experiences in protein isolation techniques, the aim of the present study was focused on the preparation of hemp protein isolate (HPI) with low-cost technology; furthermore, we also evaluated the quality of isolated protein by studying its physicochemical and functional properties to assess its potential application in food industry.

## 2. Results

### 2.1. Proximate Analysis of Hemp Seeds

The proximate analysis of hemp seeds is presented in (Table 1). The moisture content was 10.00 ± 0.33%, ash content was 4.00 ± 0.08%, crud fiber content was 12.00 ± 0.51%, crud protein content was 21.00 ± 0.04%, crud fat content was 28.00 ± 1.35% and total carbohydrate content was 25.00 ± 1.39%. The proximate analysis of the hemp seed in the current study showed that it is a high nutritional quality and potentiality as a nutritious and health-promoting food due to its content of essential nutrients such as protein, carbohydrate, fat, and fiber. The findings of the present investigation were in harmony with the previous research results, which emphasized the high-protein content of hemp seeds (20–25%). Furthermore, we found that hemp protein is easy to digest and contains all essential amino acids [7,21]. Hemp seed might be claimed as a good alternative plant-based food due to its content of essential macro -and- micronutrients.

### 2.2. Hemp Seeds Protein Isolate (HPI)

The extractability of protein from hemp seeds at variable pH values (2–12) has been shown in Figure 1a. The results showed that the best pH value for extracting the most proteins was the alkaline pH (10–12). Notably, the extractability of the protein was significantly increased by increasing the pH value up to 12 (49.10 ± 0.12 mg/mL) (*p* ≤ 0.05). The significant increasing in protein extractability was started from pH 10 to 12 (35.65 ± 0.78 to 49.10 ± 0.12 mg/mL) (*p* ≤ 0.05) (Figure 1a). The lowest extractability was noted on the pH around the isoelectric point [4,5] (5.33 ± 0.51 and 7.68 ± 0.38). The solubility of protein on the alkaline pH values emphasized the high content of acidic amino acids that ionized at the alkaline pHs. The previous work of [19,20,22] stated that the alkaline pH is suitable for extracting the plant protein content while the extracted proteins could be recovered at the acidic pH (4.6) as the isoelectric point. pH is the impacting optimum factor in the extraction of hemp protein. At the acidic pH (2 to 7), the extraction yields are low and quite constant, corresponding mainly to hemp albumins solubilization. The extraction of globulin starts to increase significantly from pH 8. Aqueous extraction at strong alkaline pH (>9) without salt addition should be retained to optimize the protein extraction yield [23]. On the other hand, the protein was precipitated from protein solution at the acidic pH value (4.6) (isoelectric point), which showed the best recovery of the hemp proteins (78.4 ± 0.13% by Kjeldahl) (data are not tabulated). The recovery of protein at the isoelectric point of the hemp seed was previously reported in several studies. The recovery of hemp seed protein at the isoelectric point (pH 4.5) was 84% by [24] and 50.60% of protein recovered by [22]. The differences in protein recovery might be referred to several factors such as the variety of the seeds, the protein–protein interaction that affected the level of protein recovery, and the isolation conditions as well. 

### 2.3. Characterization of Hemp Protein Isolate (HPI)

#### 2.3.1. Hemp Protein Solubility

Protein solubility is defined as the concentration of protein in a saturated solution that is in equilibrium with a solid phase, under a set of conditions. Solubility can be influenced by several extrinsic and intrinsic factors [25]. Protein solubility of hemp protein isolate at various pH values is shown in Figure 1b. The protein solubility was gradually increased with increasing the pH values. The highest solubility degree was noted at the pH values of 11 and 12 (64.05 ± 0.26–69.10 ± 0.35%), while the lowest protein solubility was recorded at pH 4.6. The degree of solubility of the protein might be according to the degree of denaturation and the type of amino acids. The behavior of the hemp protein isolate is similar to most previously studied plant proteins such as soy, chickpea, lentil, lupin, and quinoa proteins, which showed the maximum solubility at alkaline pH value [9,10,11,12,19,20,26]. It is often difficult to accurately determine the solubility of protein isolate, due to the forming of gel-like or supersaturated solutions. When lyophilized protein is added to solvent, the variable water and salt content of the lyophilized powder is difficult to control and can have a significant effect on solubility measurements [25].

#### 2.3.2. Electrophoresis Analysis of Hemp Protein Isolate (HPI)

The SDS–PAGE profile of the HPI is presented in Figure 1 c, d. The most predominant protein appeared in the band with high intensity and MW 58.1 KDa, which may be corresponding to edestin [27] followed by the protein with a molecular weight 31.5 KDa, which may be attributed to albumin [6] (Figure 1d. On the other hand, the HPI electrophoresis profile showed another four bands with moderate concentrations according to the intensity of the bands (Figure 1d) with the molecular weights 63.2 KDa, 53.2 kDa, 49.8 KDa and 41.4 kDa. Globulin protein is about 85% of total hemp proteins [28], which exhibited the lowest solubility at pH 5 [24]. The globulin fraction is made of 93% of 11 s legumin, which is called edestin [22,23,28]. Edestin is a hexamer with a molecular weight of about 300 KDa (MW of monomer about 50 KDa) linked by non-covalent interactions [22,23,28].

#### 2.3.3. Secondary Structure of HPI

Fourier transform IR (FTIR) spectroscopy is a widely used technique for characterizing the structure of proteins and peptides. Proteins and peptides mainly contain three absorption bands: amide I band, amide II band, and amide III band. The most sensitive spectral region of the protein secondary structural component is the amide I, which might be detected at 1700 to 1600 cm^−1^, and which is due almost entirely to the C=O stretch vibrations of the peptide linkages [29,30]. The amide I band spectrum in the wavelengths ranging from 1700 to 1600 cm^−1^, which covers the typical amide I peak, was useful for the analysis of protein secondary structural composition and conformational changes [31]. The FTIR spectroscopy of the hemp protein isolate (HPI) is presented in Figure 2. The absorption peaks of the amide I band were shown in 1750 and 1600 cm^−1^, which pointed to C=O stretching and N-H stretching at 1650–1580 cm^−1^. In addition, the third peak at 3250 related to N-H stretching for the aliphatic primary amine around 3400–3300 cm^−1^ and the N-H stretching for secondary (II) amine appeared at 3350–3310 cm^−1^ (Figure 2). The different regions of amide I were assigned to specific protein secondary structures with β-sheet at 1612–1625 cm^−1^, 1625–1640 cm^−1^, ɑ-helix at 1650–1660 cm^−1^, and random coil at 1637–1645 cm^−1^ [32].

#### 2.3.4. Amino Acids Composition of Hemp Protein Isolate (HPI)

The amino acids profile of the HPI in the current study emphasized that hemp protein contains most of the essential amino acids that are required by the human body (Table 2). The hemp protein isolate (HPI) showed a high content of arginine 15.52 g/100 g, phenylalanine+ tyrosine 9.63 g/100 g, methionine + cysteine 5.49 g/100 g, leucine + isoleucine 5.21 g/100 g, and valine 4.53 g/100 g. On the other hand, the HPI contains a moderate level of threonine 3.29 g/100 g and lysine 2.50 g/100 g, with a low level of tryptophan 0.22 g/100 g as the limiting amino acid. The level of lysine and sulphur amino acids in hemp protein isolate are significantly higher than those found in most plant proteins, which usually are poor in lysine and sulphur-containing amino acids. On the other hand, the HPI still has insufficient lysine content to meet the recommended daily requirements (Table 2). Generally, most essential amino acid levels were satisfactory for the FAO/WHO daily requirements. Hemp protein isolate (HPI) previously was characterized as a rich source of glutamine, arginine, tyrosine, and phenylalanine, while it is limited in lysin, sulfur-containing amino acids, and tryptophan [1,33,34]. The digestibility and availability of hemp seed protein were higher than other plant proteins such as wheat and legumes [34]. The application of hemp protein isolates in food processing needs the addition of tryptophan and lysine for the fortification of food products to meet the human daily requirements of essential amino acids. 

#### 2.3.5. Water-and-Oil Holding Capacity (WHC and OHC)

The water-and-oil holding capacity is a very important property in food processing where it plays a crucial role in the texture, taste, and organoleptic properties of the products. In the present study, the HPI showed the highest-water holding capacity value (4.21 ± 0.24 mL/g protein) at pH 5, which is close to the isoelectric point (the precipitating pH value) and thus makes it suitable for different food products, especially bakery products and viscous foods (Table 3). The function of the WHC of protein isolate in the food system is dependent on its water-binding ability, which is affected by its solubility, pH and ionic strength [6]. The interaction of the amino group (NH_3_) and carboxyl group (COO^−^) of protein molecules with water may increase the solubility of protein [21]. On the other hand, the HPI showed an oil-holding capacity of 2.56 ± 0.08 mL/g protein at pH 5 (Table 3). The oil-holding capacity of hemp protein isolate might be due to its high-protein content, which enhances the interactions with the lipid phase more than polysaccharides. This phenomenon makes the protein more able to form the emulsion and be more suitable for the processing of different foods. The oil-holding capacity of freeze-dried hemp protein isolate was previously noted as 1.79 mL oil/g protein [35]. Gulsah and Oktay [36] reported that the WHC of hemp protein isolate was 1.20 mL/g and the OHC was 1.72 mL/g. Consequently, it could be said that the HPI showed a WOC and OHC appropriate for food applications.

#### 2.3.6. Foaming Activity and Stability

The foaming ability of food ingredients plays a key role in the flavor quality and appearance characteristics of food. In the present study, the foaming ability and stability were studied at different pH values (2–11) (Table 3). The foaming activity of the HPI was significantly increased with increasing the pH from 59.46 ± 2.58% at pH 2 to 67.23 ± 3.20% at pH 11 (*p* ≤ 0.05). The maximum foaming capacity was noted at pH 11, which meets the maximum solubility of the protein and emphasizes the direct relation between the foaming capacity and protein concentration. The findings of the current investigation are in line with Britten and Lavoie [37], who stated that the foaming capacity of protein was increased with increasing the protein concentration up to 2%. The minimum foaming capacity of the HPI was noted at pH 5.0, which is close to the isoelectric point of proteins (pH 4.5–5.0) [38]. The same trend was noted with the foaming stability, where the foaming stability was increased with increasing the pH values and reached the maximum stability at pH 11, where there was the maximum solubility of the protein isolate. Generally, the HPI showed a significant foaming capacity and stability, which pointed to the suitability of the HPI for different food application.

#### 2.3.7. Emulsion Activity Index and Emulsion Stability Index

The emulsion capacity and stability of the HPI were evaluated because the emulsions confer to food some distinct functional attributes, such as desirable appearances, textures, mouth feeling, and flavor profiles. Moreover, emulsions are a widely used vehicle for the encapsulation and delivery of bioactive compounds. The emulsion capacity and stability of the HPI were significantly increased with increasing the pH value (*p* ≤ 0.05). The highest EAI was noted at pH 9 (91.3 ± 2.57 m^2^/g) with low stability (19.15 ± 2.03) (Table 3). On the other hand, the highest stability index was at pH 3 (49 ± 5.05). The lowest EAI was noted at pH values of 4, 5 and 6 with no significant differences between them (*p* ≥ 0.05) (Table 3). There was a direct relation between emulsion capacity and protein solubility, where the lowest solubility was determined at pH values close to the isoelectric point of the protein, while the highest emulsion ability was noted at the alkaline pH values where the protein showed a high solubility. The variation of ESI was related to the diffusion coefficient [35]. The high emulsifying capacity in pH 9 indicated that the higher protein solubility resulted in faster protein adsorption to the water–oil interface with more flexible and mobile structures. This finding was in contrast with previous work that reported that the high EAI of the HPI was noted at a neutral pH value (pH 7.0) [39]. On the other hand, the findings of the current work agreed with [38], which reported the lowest emulsion ability index of the HPI at pH 7. Obviously, the HPI showed a good emulsion capacity and stability, which emphasizes its potential application in food processing.

#### 2.3.8. Correlation Coefficient

The analysis of data to find the relation coefficient between the different parameters in Table 4 showed a strong linear positive correlation (r = 0.623) between protein concentration and protein solubility. In the same trend, a very strong linear positive correlation (r = 0.980) was noted between the foaming capacity and the stability. On the other hand, a medium negative correlation (r = −0.504) was remarked between the emulsion ability index and the emulsion stability index.

## 3. Materials and Methods

### 3.1. Materials

Hempseeds (*Cannabis sativa* L.) were purchased from local market in Chelyabinsk, Russia. Chemicals and reagents were purchased from Sigma-Aldrich.

### 3.2. General Analysis of Hempseeds

Moisture, total protein, total lipids, total fiber and ash were determined according to Siano et al. [40]. Total carbohydrate content was determined by differences according to the following equation:***Total carbohydrates (%) = 100 – (moisture + total protein + total lipids + total fiber + ash)***(1)

### 3.3. Preparation of Hemp Protein Isolate

In the current study, protein isolate was prepared through the main five steps: (i) removing of fat content; (ii) extraction of protein from the seeds; (iii) protein precipitation; (iv) protein neutralization and (v) protein lyophilization [19,20].

#### 3.3.1. Removing of Fat Content (Defatting)

The defatting of hempseed samples was conducted by homogenizing the samples (50 g seeds’ powder) with chloroform/methanol (2:1, *v*/*v*) (1000 mL) following the method of Bligh and Dyer, 1959 [41]. The final volume was 20-fold the sample volume (1 g in 20 mL of solvent mixture). After dispersion, the whole mixture was shaken for 120 min in a shaker (Solaris-2000 Thermo Fisher Scientific, Paisley, UK) at room temperature. Then, the homogenate was filtered with filter paper (Wattman # 1) to recover the liquid phase. This step was repeated three times to extract all lipids in the sample. Finally, the defatted hempseed flour was dried at 45 °C in a hot air flow.

#### 3.3.2. Protein Extraction

Defatted hemp seed flour (50 g) was suspended in 1000 mL of deionized distilled water (ddH_2_O) (1:20, *w*/*v*), and the pH of the solution was adjusted to 11.0 using 0.5 molL^−1^ NaOH. The suspensions were stirred on a magnetic stirrer (MS-500P-SET-INTLLAB, SHENZHEN, China) for 1 h to extract the protein and the pH of the mixture was checked every 15 min to be maintained at pH 11. After extraction, the samples were centrifuged at 7000× *g* for 30 min at 20 °C and the supernatant containing the protein (protein solution) was collected.

#### 3.3.3. Protein Precipitation

The precipitation of protein from the supernatant was carried out by adjusting the pH at 4.6 (isoelectric point) with 0.5 molL^−1^ HCl. The precipitate was then collected by centrifugation at 12,000× *g* at 4 °C for 45 min. The protein isolates were collected, neutralized, and lyophilized.

### 3.4. Physicochemical Properties of Protein Isolate

#### 3.4.1. Protein Solubility

The solubility of hempseed protein isolate (5% suspension) was determined at pH values ranging from 2 to 12 according to Potin et al. [23]. For a better solubilization, the suspensions were stirred at room temperature (25 °C ± 2) for 1 hr on a magnetic stirrer. The pH values were adjusted using (0.1 molL^−1^) HCl and (0.1 molL^−1^) NaOH. Then the suspensions of all tested pH values were centrifuged at 7000× *g* (Centrifuge model K241R, Centurion Scientific Ltd., West Sussex, PO18 9JL, UK). Total protein content in the supernatant was determined by the Bradford method. The experiment was conducted in triplicate.

#### 3.4.2. Amino Acid Composition

Amino acid composition of the HPI was determined by capillary electrophoresis (Capel-105M, Lumex, Saint-Petersburg, Russia) according to Latorre et al. [42]. The HPI sample (250 μL) was mixed with 250 μL of NQS solution (1,2-Naphthoquinone-4-Sulphonic Acid Sodium Salt) and 250 μL of borate buffer solution was added. The mixture was placed in a water bath at 65 °C for 5 min after adjusting the pH at 10, and 250 μL citrate buffer was added for stopping the reaction and stabilizing the derivatives. The mixture was filtered through a nylon membrane (0.45 μm) and injected to the system. The separation mobile phase was 40 mM of sodium tetraborate solution (pH 9.2) and isopropanol (3:1, *v*/*v*). The capillaries were flushed with 0.1 molL^−1^ NaOH for 20 min. The amino acids were separated by cation exchange column (200 × 4-mm, 8-μm particle size) using lithium solutions.

#### 3.4.3. Sodium Dodecyl Sulfate–Polyacrylamide Gel Electrophoresis (SDS–PAGE)

Sodium Dodecyl Sulfate–Polyacrylamide Gel Electrophoresis (SDS–PAGE) of hemp protein isolate (HPI) was conducted according to Laemmli [43] for protein profiling. Samples were prepared from 500 μL protein solution and were added to 1 mL buffer (distilled water, 0.5 M Tri–HCl pH 6.8, glycerol, 10% SDS, 1% bromophenol blue and β-mercaptoethanol) and heated at 98 °C for 10 min, then applied to the sample wells. The sample (20 μL) and 5 μL of the protein marker (200, 150, 100, 85, 60, 50, 40, 30 and 25 KDa) were loaded on the gel. Electrophoretic migration was conducted at constant current (14 mA/gel) for 60 min. Gel was fixed with fixing solution (water/methanol/acetic acid 700 mL:200 mL:100 mL *v*:*v*:*v*) for 30 min and then stained with Coomassie Brilliant Blue (R-250) for 1 hr. The stained gel was de-stained by frequently changing the fixing solution until the excess stain disappeared.

#### 3.4.4. Secondary Structure of Protein (FTIR)

FTIR spectroscopy of the samples was recorded using a WQF-510 FTIR spectrometer (BFRL, Beijing, China) at a wavelength ranging from 4000–400 cm^−1^ under ambient conditions [29]. Each spectrum was collected by averaging 64 scans with a resolution of 4 cm^−1^. The 1–2 mg dried HPI samples were blended with 400 mg KBr and pressed into pellets before spectrum acquisition. The original FTIR spectrum was smoothed for further deconvolution analysis with the aid of Peak Fit v4.12 (SeaSolve Softwave Inc., Framingham, MA, USA) software. Each measurement was carried out in triplicate.

### 3.5. Functional Properties of Protein Isolate

#### 3.5.1. Water-and-Oil Holding Capacity

The water-and-oil holding capacity of hemp protein isolate (HPI) was performed by following [20,44]. The HPI (500 mg) was mixed with 10 mL of distilled water/sunflower oil and allowed to stand at ambient temperature (30 ± 2 °C) for 30 min, then the mixture was centrifuged at 3500× *g* for 30 min. The water-and-oil holding capacity in (mLg^−1^) was calculated according to the following equation:**(W/O) HC = V _water (oil) initial_ − V _supernatant_**
where: (**W/O**) **HC**: water- or oil-holding capacity; **V**: volume of water or oil.

#### 3.5.2. Foaming Capacity (FC) and Stability (FS)

**FC** and **FS** were carried out following the method of El-Sohaimy et al. [20]. The protein solution was agitated in a blender (Fimar FR200P, Rome, Italy) at high speed for 5 min and then transferred into graduated cylinders. The pH values varied from 2.0 to 12.0 using (0.1 molL^−1^) HCl and (0.1 molL^−1^) NaOH. The foam capacity was calculated according to the following equation:**% FC = (V _after agitation_ − V _prior agitation_/V _prior agitation_) × 100**
where: **FC**: foaming capacity.

Similarly, the FS value was determined, but the samples were allowed to stand at room temperature for 30 min, and the residual foam volume (**V _residual foam_**) was calculated according to the following equation:**% FS= (V _residual foam_/V _total foam_) × 100**
where: **% FS**: foaming stability.

#### 3.5.3. Emulsifying Activity Index (EAI) and Emulsion Stability Index (ESI)

The **EAI** and **ESI** were evaluated by following the method of [45]. The protein solution (15 mL) was mixed with 5 mL of sunflower oil. The pH values of the mixtures ranged from 2.0 to 12.0. The mixtures were homogenized at 7500× *g* for 1 min, using a homogenizer (Homelab, Shanghai, China). Then, 50 µL aliquots were taken from the emulsions at 0 and 10 min from the bottom of the tube and mixed with 10 mL of 0.1% sodium dodecyl sulphate (SDS) (1:200 dilution). The absorbance of the diluted solutions was noted at 500 nm immediately after emulsion formation (**A_0_**) and at 10 min (**A_10_**). The **EAI** and **ESI** were calculated using the following equations:**EAI (m^2^/g) = 2T × F × A_0_/C × θ × 10**
**ESI: {min****θ_0_ (∆_t_/∆_A_)}**
$$\Delta A=A_{0}-A_{10}\;\mathrm{and}\;\Delta t=10\;\min$$
where **EAI**: emulsion ability index; **ESI**: emulsion stability index; **T** = 2.303; **F**: dilution factor (200); **A_0_**: absorbance measured at 500 nm immediately after emulsion formation; **A_10_**: absorbance measured at 500 nm after 10 min of emulsion formation **c**: protein concentration (0.01 g/mL) and **θ**: dispersed phase (oil) volume fraction [15].

### 3.6. Statistical Analysis

Data were expressed as mean ± standard error of the mean by the multiple comparison one-way analysis of variance (ANOVA) and Tukey’s test using the SPSS16 software program with probability (*p*)-values < 0.05 considered statistically significant. All experiments performed in three replicates.

## 4. Conclusions

The preparation of hemp protein isolate can be performed by following the easy-to-use and costless procedure by extracting the protein at the alkaline pH (pH 10–12) and precipitating it at pH (4.5–4.6) as an isoelectric point. Edastin is the most predominant protein in the protein profile with a molecular weight of 58.1 KDa beside albumin with a molecular weight of 31.5 KDa. The HPI contained an adequate amount or concentrationof essential amino acids such as arginine (15.52 g/100 g), phenylalanine + tyrosine (9.63 g/100 g), methionine + cysteine (5.49 g/100 g), leucine + isoleucine (5.21 g/100 g), and valine (4.53 g/100 g) with limiting amino acid of tryptophan (0.22 g/100 g HPI). The HPI showed a good water-and-oil holding capacity close to the PI point. The highest foaming capacity was noted at pH 11 (67.23 ± 3.20%). while the highest emulsion ability index was noted at pH 9 (91.3 ± 2.57 m^2^/g). It could be concluded that the hemp protein isolate (HPI) of the variety “*Cannabis sativa ssp. Sativs*” showed adequate nutritional and functional properties that encourage food technologists to use it as a promising alternative protein source for developing/ formulation of rich protein foods and fortification of some food products with a good protein isolate for improving their nutritional quality and technological properties. Enrichment of beverages with proteins has become a major trend in the market. At the same time, the beverage industry is seeking protein-rich products because they are preferable for consumers. Consequently, the current study is continuing to utilize the HPI for developing high-protein beverages, such as fermented beverages, sports drinks, soft drinks, and juices by using appropriate technologies in the production process.

## Figures and Tables

**Figure 1 plants-11-02825-f001:**
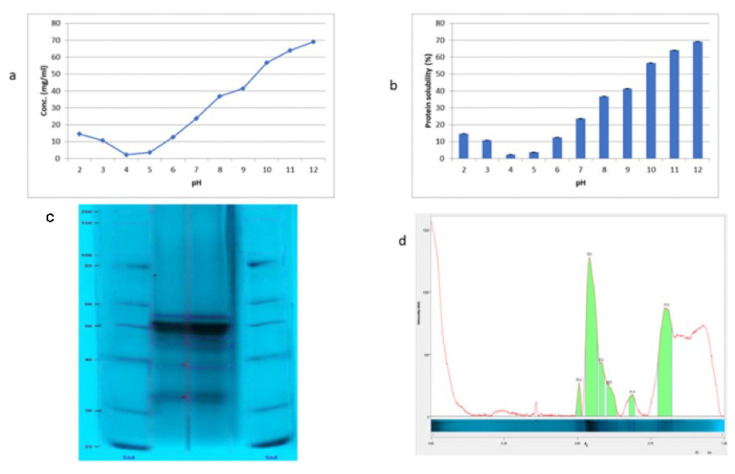
(**a**). Extractability of hemp seeds protein at different pH values, (**b**). Protein isolate solubility power at different pH values, (**c**). SDS–PAGE protein profile, (**d**). the intensity of protein bands of hemp protein isolate (HPI). Values shown in means ± SD.

**Figure 2 plants-11-02825-f002:**
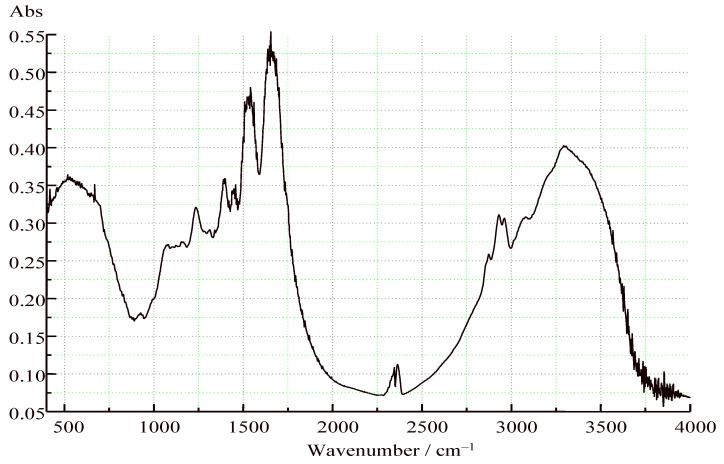
FTIR spectrum of hemp protein isolate (HPI).

**Table 1 plants-11-02825-t001:** Chemical composition of hemp seeds.

Parameter of Analysis	Content (%)
Total Protein	21.00 ± 0.04
Total lipid	28.00 ± 1.35
Crude Fiber	12.00 ± 0.51
Ash	4.00 ± 0.08
Moisture	10.00 ± 0.33
Total carbohydrates	25.00 ± 1.39

**Table 2 plants-11-02825-t002:** Amino acids composition of hemp protein isolate (HPI).

Essential Amino Acid	Content, g/100 g	AAS *	FAO/WHO Pattern for Adults	Satisfaction of Daily Requirement, %
Isoleucine + Leucine	5.21	47.00	3.20	141.60
Lysine	2.88	52.00	1.60	156.60
Methionine + Cysteine	5.49	157.00	1.70	280.70
Phenylalanine + Tyrosine	9.63	161.00	1.90	440.30
Threonine	3.79	95.00	0.90	365.80
Tryptophan	0.26	26.00	0.50	44.40
Arginine	15.52		0.46	
Valine	4.53	91.00	1.80	218.70
Total of essential amino acids	31.79	–		
Non-Essential amino acids				
Arginine	15.52		0.46	
Histidine	3.20		1.60	
Proline	3.44		0.61	
Serine	4.05		0.53	
Alanine	3.85		0.26	
Glycine	3.70		0.20	
Glutamic acid + glutamine	3.91		1.75	
Asparagine + aspartic acid	12.53		0.88	

* AAS= Amino acid score.

**Table 3 plants-11-02825-t003:** Functional properties of HPI.

pH	WOC (mL/g)	OHC (mL/g)	Foaming Capacity (%)	Foaming Stability (%)	EAI (m^2^/g)	ESI (min)
2	2.10 ± 0.13 ^a^	1.23 ± 0.17 ^a^	59.46 ± 2.58 ^b^	53.43 ± 3.41 ^a^	38.89 ± 2.47 ^f^	27.05 ± 2.22 ^bc^
5	4.21 ± 0.24 ^b^	2.56 ± 0.08 ^b^	47.93 ± 3.01 ^c^	39.50 ± 2.82 ^b^	25.48 ± 0.30 ^g^	49.00 ± 5.05 ^a^
7	3.52 ± 0.21 ^b^	2.31 ± 0.16 ^b^	20.26 ± 0.37 ^d^	14.90 ± 2.07 ^c^	26.82 ± 0.57 ^g^	44.86 ± 6.72 ^a^
9	2.81 ± 0.15 ^a^	1.36 ± 0.14 ^a^	44.73 ± 2.49 ^c^	33.40 ± 3.02 ^b^	27.43 ± 0.29 ^g^	24.10 ± 1.92 ^bcd^
11	2.36 ± 0.12 ^a^	1.23 ± 0.21 ^a^	67.23 ± 3.20 ^a^	54.73 ± 3.34 ^a^	20.47 ± 0.76 ^g^	21.77 ± 1.36 ^bcde^

Values with different letters in the same column indicate significant differences (*p* ≤ 0.05).

**Table 4 plants-11-02825-t004:** Correlation between protein concentration and solubility, foaming capacity and stability, emulsion capacity and stability.

Relationship	Correlation Coefficient (r)	*p* (2-Tailed)
Concentration of protein and Solubility of protein	0.623 **	0.000
Foaming capacity and Foaming stability	0.980 **	0.000
EAI and ESI	−0.504 **	0.005

r = 1(perfect positive correlation); r = 1–0.5 (strong positive correlation); r = 0 (no correlation); r= −1 (strong negative correlation); **- strong correlation.

## Data Availability

Not applicable.

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
