# Peer review of "Nutritional Quality, Chemical, and Functional Characteristics of Hemp (Cannabis sativa ssp. sativa) Protein Isolate"

_plants, 2022, doi:10.3390/plants11212825_

Round 1

Reviewer 1 Report

The article presents the methods of obtaining the isolate as an ecological and industrially cheap method, but this method has been used for many years, there is nothing new in it. Check the bibliography please.

Line 30-31 Why triptophan expressed relative to 100 grams of HPI?

Line 42-43 "Hemp (Cannabis sativa ssp. satvs) is one of the oldest plants cultivated in Asian countryes more than 5 centuries ago" Are you sure? Check the bibliography please.

References to obtaining protein isolates carried out by other research groups should appear (see references 19 and 20)

line 74, 229, 231, 305 references appear in two differents formats

There is no agreement in the way of expressing the results numerically, sometimes decimals are missing (lines 115-118).

Lines 294-297 say that the percentage of carbohydrates is calculated by subtracting the percentage of the other compounds from 100, so it is to be expected that the sum of all the components equals 100, but this is not the case, since in table 1 the sum of the percentages is 118.

line 541 number 45 is missing

Author Response

Response to reviewer #1 comments

First of all, I would like to thank a reviewer for his time and efforts to improve the quality of the manuscript. It is my pleasure to seriously consider the comments point by point in the manuscript. All responses to the comments have been highlighted with yellow color throughout the manuscript.

# The article presents the methods of obtaining the isolate as an ecological and industrially cheap method, but this method has been used for many years, there is nothing new in it. Check the bibliography please.

Response, I completely agree that is not a new method for preparation a plant protein isolate as I have a pervious article on the preparation of protein isolate from Chickpea, lupin, lentil and quinoa (El-Sohaimy, S. A.; Refaay T. M.; Zaytoun M. A. M. Physicochemical and functional properties of quinoa protein isolate. Ann. Agric. Sci., 2015, 60(2), 297–305. http://dx.doi.org/10.1016/j.aoas.2015.10.007, El-Sohaimy, S. A.; Brennan M. A.; Darwish A. M. G.; Brennan C. S. Chickpea Protein Isolation, Characterization and Application in Muffin Enrichment. International Journal of Food Studies, 2021. 10, SI57-SI71. DOI: 10.7455/ijfs/10.SI.2021.a5 and el-Sohaimy S.A, Sitouhy M and El-Masry R. 2007. Isolation and Partial Characterization of Chickpea, lupine and Lentil seed proteins. World Journal of Agricultural Sciences, 3(1), 123-12.9.). So, the current manuscript used this method as a cheapest methodology for industry and ecofriendly for preparing and characterizing the hemp protein isolate to evaluate its quality and suitability for food industry.

# Line 30-31 Why triptophan expressed relative to 100 grams of HPI?

Response: The measuring unit of all amino acids expressed in g/100g not only tryptophan. So, we cannot express all amino acids in g/100g and tryptophan alone in (mg/100g or mg/g).

# Line 42-43 "Hemp (Cannabis sativa ssp. satvs) is one of the oldest plants cultivated in Asian countryes more than 5 centuries ago" Are you sure? Check the bibliography please.

Response: Changed to “Hemp (Cannabis sativa ssp. sativs) is one of the oldest plants cultivated in Asian countries.”

“John M. McPartland,William Hegman, Tengwen Long. Cannabis in Asia: its center of origin and early cultivation, based on a synthesis of subfossil pollen and archaeobotanical studies. Vegetation History and Archaeobotany (2019) 28:691–702 https://doi.org/10.1007/s00334-019-00731-8”

# References to obtaining protein isolates carried out by other research groups should appear (see references 19 and 20)

Response: Other research groups who are previously worked on HPI were already mentioned and discussed throughout the manuscript.

For instance:

(1) Leonard, W.; Zhang, P.; Ying, D.; Fang, Z. Hempseed in food industry: nutritional value, health benefits, and industrial applications. Compr. Rev. Food Sci. Food Saf., 2020, 19 (1), 282-308. DOI: 10.1111/1541-4337.12517

(5) Aiello, G.; Fasoli, E.; Boschin, G.; Lammi, C.; Zanoni, C.; Citterio, A.; Arnoldi, A. Proteomic characterization of hempseed (Cannabis sativa L.). Journal of Proteomics, 2016, 147, 187– 196. https://doi.org/10.1016/j.jprot.2016.05.033/

(6) Malomo, S. A.; Aluko, R. E. A comparative study of the structural and functional properties of isolated hemp seed (Cannabis sativa L.) albumin and globulin fractions. Food Hydrocolloids, 2015, 43, 743-752. https://doi.org/10.1016/j.foodhyd.2014.08.001

(17) Korus, J. M.; Witczak, R.; Ziobro, L. J. Hemp (Cannabis sativa subsp. sativa) flour and protein preparation as natural nutrients and structure forming agents in starch-based gluten free bread. LWT - Food Sci. Technol., 2017, 84.143-150. DOI:10.1016/J.LWT.2017.05.046

# line 74, 229, 231, 305 references appear in two differents formats

Response: The citation of references in the text have been corrected accordingly

# There is no agreement in the way of expressing the results numerically, sometimes decimals are missing (lines 115-118).

Response: The mistakes have been fixed and marked in yellow color

# Lines 294-297 say that the percentage of carbohydrates is calculated by subtracting the percentage of the other compounds from 100, so it is to be expected that the sum of all the components equals 100, but this is not the case, since in table 1 the sum of the percentages is 118.

Response: the values in the table have been corrected accordingly

line 541 number 45 is missing

Response: The reference number (45) has been fixed

Reviewer 2 Report

I have attached an edited version of the paper.  Some major issues.  Only one cultivar of hemp was used for the study and there is a great deal of variation amongst varieties, so hard to suggest that the observations from this study are for all hemp. 

There is a tendency to overstate how good hemp is rather than just provide the data with a summary at the end. 

Author Response

Response to reviewer #2 comments

Firstly, I would express my deepest thank to the respected reviewer for the kind effort and time to improve the quality of the manuscript. all comments have been seriously considered point by point throughout the manuscript.

#I have attached an edited version of the paper. 

Response: All comments in attached file have been considered accordingly throughout the manuscript and highlighted with a yellow color

# Some major issues.  Only one cultivar of hemp was used for the study and there is a great deal of variation amongst varieties, so hard to suggest that the observations from this study are for all hemp. 

Response: I am completely agreeing that there is a variation among different varieties. It is according to the differences in the environmental conditions, cultivation conditions, genetic properties of the variety, and others, but in the current study we are focused on the Cannabis sativa ssp. Sativs that cultivating in the Russian territory to evaluate its ability for using in the food applications because our work is already continued for utilizing the protein isolate of this variety for development some protein-rich beverages and bakery products and also developing a protein bars for sport nutrition.

# There is a tendency to overstate how good hemp is rather than just provide the data with a summary at the end.

Response:

The summary has been improved accordingly. In the summary we have just stated that the nutritional and functional properties of the HPI are suitable for food application or not based on the obtained results. 

# In the introduction “Hemp seeds contain about 18%-23% protein”, this is not high compared to soybean.

Response: In the current study there is so comparison between Hemp seed and soybean. It is true that is the protein content of hemp seed lower than soybean, but the quality of protein is better that soybean in the term of amino acids composition and the protein digestibility and availability and also in the term of functional properties which important for utilizing it in food application.

Reviewer 3 Report

Article Plants-1981767 is of interest.

The work is well drafted and properly structured. Figures and Tables are fine. English needs further improvement, some typos have been highlighted below

I think this work fits the targets of the Journal and therefore I recommend its publication after minor revisions on Plants.

Some revisions for the authors are detailed in the list given below:

First page: remove “.” after the first author

Title: correct “sativs” in “sativa”

Line 10: put “institute” in capital letter

Line 12: “energy” is written incorrectly

Line 16: remove “a” before “rich”

Line 19: the main verb is missing

Line 24-25: write the sentence better

Line 28: remove “was” before “showed”

Line 32: remove “with” before “increasing”

Line 34: put number 4 before “conclusion”

Line 36: write “provides” and not “provide”

Line 38 and 42: correct “sativs” in “sativa”

Line 49: replace “Yso” with “Uso”

Line 56: add “and” between “edestin” and “globulin”

Line 66: add “of” between “shell” and “hemp”

Line 68: replace “format” with “form”

Line 74: remove the reference and leave only the associated number as in the other cases

Line 77: add “is” to “utilized”

Line 92: replace “targeted” with “targeting”

Line 93: put “.” and a space after “proteins”

Line 101: “for” is written in a bad way

Line 103: “from” is written in a bad way

Line 106: add a space between “pH” and “11”

Line 11-112: the main verb is missing

Line 115: add “is” to “presented”

Line 116-117 and in table 1: “crude” is written in a bad way

Line 118-120: there are several mistakes in the sentence, write it better

Line 120: in “the” the initial letter is missing

Line 122: put the subject

Line 124: “plant” is written in a bad way

In paragraph 2.1 the unit of measure of the first values is not specified

Line 136: replace “extraction” with “extracting”

Line 149: replace “affecting” with “affected”

Line 156: replace “are” with “is”

Line 161: remove “which”

Line 172: remove “was” before “showed”

Line 173: remove brackets which include the molecular weights values

Line 176: add “is” before “called” and “a” before “hexamer”

Line 183: add “a” before “widely”

Line 187: remove brackets including values

Line 192: replace “was” with ”were”

Line 196: replace “where” with “with” or add a verb

Line 196-197: remove brackets including values

Line 201-202: write the sentence better

Line 203: remove “was” before “showed”

Line 206: better to write “with tryptophan as the limiting amino acid

Line 209: add “are” before “poor”

Line 210: write better the last part of the sentence (very long sentence and the subject is missing)

Line 212: add “was” to “characterized”

Line 215: “is” is missing twice

Line 217-220: write the sentence better

Line 225-226: the verb is missing in the second part of the sentence

Line 229: put “is” instead of “are”

Line 243: put “were” instead of “was”

Line 248-249: replace “invistigation” with “investigation”

Line 252-254: write the sentence better

Line 265: put “on” in capital letter

Line 266-267: “pH values OF 4, 5 and 6…”

Line 277: remove “is” and replace “emphasize” with “emphasizes”

Line 281: remove brackets to (4); add “linear” between “strong” and “positive”

Line 282: remove the second part of the sentence from “which” to the end of the sentence; put “in” in capital letter

Line 286: no capital letters for “Foaming” and ”Emulsion”

Line 288: no capital letter for “Perfect”

Line 297: remove (1) from the equation

Line 301: A dot is missing

Line 304: replace (2:1v/v) with (2:1, v/v)

Line 305: remove brackets from (Bligh & Dyer, 1959)

Line 315-316: change the sentence. Better writing: “…to extract the protein and pH of the mixture was checked every 15 min to be maintained at pH 11”

Line 317: replace “which contain” with “containing”

Line 320: replace “adjusted” with “adjusting”

Line 322: replace “12000 x g” with “12000xg”

Line 327: remove brackets from (Klompong et al., 2007) and add the number of the reference; add “a” between “for” and “better”

Line 330: replace “7000 Xg” with “7000xg”

Line 338: remove the dot after “min”

Line 341-342: write the sentence better

Line 342: put “the” in capital letter

Line 343: remove the dot after “min”

Line 347-348: write the sentence better

Line 371: replace “Equation (2)” with “equation 2”

Line 371: remove (2) from the equation

Line 378: no capital letter for “Equation”

Line 378: remove (3) from the equation

Line 382: no capital letter for “Equation”

Line 382: remove (4) from the equation

Line 385-386: replace “ml” with “mL”

Line 387: replace “7000 Xg” with “7000xg”

Line 391: no capital letter for “Equation”

Line 392: remove (5) and (6) from the equations

Line 402: the dot at the end of the sentence is missing

Line 405-406: substitute “precipitate” with “precipitating”

Line 409: add space between “phenylalanine” and “+”

Line 413-417: write the sentence better, it’s way too long and complicated

Line 418-419: “beverage industry IS seeking protein-rich products because they ARE preferable for consumers”

Line 419-422: Consequently, the nutritional and functional properties of the hemp seeds have been explored in the current study might be make it MADE applicable not only to milk replacers, but also to other sorts of beverages…”

Some references should be uniformed to the others (date in bold, doi in the same format, journals in italic and abbreviated, homogenous line spacing).

Author Response

Response to reviewer #3 comments

Firstly, I would express my deepest thank to the respected reviewer for the kind effort and time to improve the quality of the manuscript. all comments have been seriously considered point by point throughout the manuscript.

# The work is well drafted and properly structured. Figures and Tables are fine. English needs further improvement, some typos have been highlighted below

Response: Thank you very much for your kind comment, The English and typewriting mistakes have been rechecked throughout the manuscript

# I think this work fits the targets of the Journal and therefore I recommend its publication after minor revisions on Plants.

Response: Thank you, the valuable comments have been seriously considered throughout the manuscript

Some revisions for the authors are detailed in the list given below:

 First page: remove “.” after the first author

Removed

Title: correct “sativs” in “sativa”

Corrected

Line 10: put “institute” in capital letter

Corrected

Line 12: “energy” is written incorrectly

Corrected

Line 16: remove “a” before “rich”

Corrected

Line 19: the main verb is missing

Corrected

Line 24-25: write the sentence better

The sentence has been rewritten accordingly

Line 28: remove “was” before “showed”

Removed

Line 32: remove “with” before “increasing”

The sentence “The foaming capacity of HPI was increased with increasing the pH values to reach the maximum value at pH 11 (67.23±3.20 %).” If the word “with has been removed the meaning of the sentence will change”

Line 34: put number 4 before “conclusion”

Corrected

Line 36: write “provides” and not “provide”

Corrected

Line 38 and 42: correct “sativs” in “sativa”

Corrected

Line 49: replace “Yso” with “Uso”

Changed

Line 56: add “and” between “edestin” and “globulin”

 The word “and” has been added

Line 66: add “of” between “shell” and “hemp”

Added

Line 68: replace “format” with “form”

Replaced

Line 74: remove the reference and leave only the associated number as in the other cases

Removed

Line 77: add “is” to “utilized”

“is” has been added

Line 92: replace “targeted” with “targeting”

Changed

Line 93: put “.” and a space after “proteins”

Changed

Line 101: “for” is written in a bad way

Corrected

Line 103: “from” is written in a bad way

Corrected

Line 106: add a space between “pH” and “11”

Space has been added

Line 11-112: the main verb is missing

Added

Line 115: add “is” to “presented”

Corrected

Line 116-117 and in table 1: “crude” is written in a bad way

Corrected

Line 118-120: there are several mistakes in the sentence, write it better

Corrected

Line 120: in “the” the initial letter is missing

Corrected

Line 122: put the subject

Corrected

Line 124: “plant” is written in a bad way

Corrected

In paragraph 2.1 the unit of measure of the first values is not specified

Corrected

Line 136: replace “extraction” with “extracting”

Corrected

Line 149: replace “affecting” with “affected”

Corrected

Line 156: replace “are” with “is”

Corrected

Line 161: remove “which”

Removed

Line 172: remove “was” before “showed”

Removed

Line 173: remove brackets which include the molecular weights values

Removed

Line 176: add “is” before “called” and “a” before “hexamer”

Added

Line 183: add “a” before “widely”

Added

Line 187: remove brackets including values

Removed

Line 192: replace “was” with ”were”

Corrected

Line 196: replace “where” with “with” or add a verb

Changed

Line 196-197: remove brackets including values

Removed

Line 201-202: write the sentence better

Re-written

Line 203: remove “was” before “showed”

Removed

Line 206: better to write “with tryptophan as the limiting amino acid

Corrected

Line 209: add “are” before “poor”

Added

Line 210: write better the last part of the sentence (very long sentence and the subject is missing)

Reformatted

Line 212: add “was” to “characterized”

Added

Line 215: “is” is missing twice

Corrected

Line 217-220: write the sentence better

Reformatted

Line 225-226: the verb is missing in the second part of the sentence

Reformatted

Line 229: put “is” instead of “are”

Replaced

Line 243: put “were” instead of “was”

Corrected

Line 248-249: replace “invistigation” with “investigation”

Corrected

Line 252-254: write the sentence better

Reformatted

Line 265: put “on” in capital letter

Corrected

Line 266-267: “pH values OF 4, 5 and 6…”

Corrected

Line 277: remove “is” and replace “emphasize” with “emphasizes”

Corrected

Line 281: remove brackets to (4); add “linear” between “strong” and “positive”

Corrected

Line 282: remove the second part of the sentence from “which” to the end of the sentence; put “in” in capital letter

Corrected

Line 286: no capital letters for “Foaming” and ”Emulsion”

Corrected

Line 288: no capital letter for “Perfect”

Corrected

Line 297: remove (1) from the equation

Removed

Line 301: A dot is missing

Added

Line 304: replace (2:1v/v) with (2:1, v/v)

Corrected

Line 305: remove brackets from (Bligh & Dyer, 1959)

Removed

Line 315-316: change the sentence. Better writing: “…to extract the protein and pH of the mixture was checked every 15 min to be maintained at pH 11”

Changed

Line 317: replace “which contain” with “containing”

Changed

Line 320: replace “adjusted” with “adjusting”

Changed

Line 322: replace “12000 x g” with “12000xg”

Corrected

Line 327: remove brackets from (Klompong et al., 2007) and add the number of the reference; add “a” between “for” and “better”

Corrected

Line 330: replace “7000 Xg” with “7000xg”

Corrected

Line 338: remove the dot after “min”

Moved

Line 341-342: write the sentence better

Reformatted

Line 342: put “the” in capital letter

Corrected

Line 343: remove the dot after “min”

Removed

Line 347-348: write the sentence better

Reformatted

Line 371: replace “Equation (2)” with “equation 2”

Corrected

Line 371: remove (2) from the equation

Corrected

Line 378: no capital letter for “Equation”

Corrected

Line 378: remove (3) from the equation

Corrected

Line 382: no capital letter for “Equation”

Corrected

Line 382: remove (4) from the equation

Corrected

Line 385-386: replace “ml” with “mL”

The mL is correct

Line 387: replace “7000 Xg” with “7000xg”

Corrected

Line 391: no capital letter for “Equation”

Corrected

Line 392: remove (5) and (6) from the equations

Corrected

Line 402: the dot at the end of the sentence is missing

Corrected

Line 405-406: substitute “precipitate” with “precipitating”

Corrected

Line 409: add space between “phenylalanine” and “+”

Added

Line 413-417: write the sentence better, it’s way too long and complicated

The Summary has been reformatted and improved

Line 418-419: “beverage industry IS seeking protein-rich products because they ARE preferable for consumers”

Corrected

Line 419-422: Consequently, the nutritional and functional properties of the hemp seeds have been explored in the current study might be make it MADE applicable not only to milk replacers, but also to other sorts of beverages…”

Corrected

Some references should be uniformed to the others (date in bold, doi in the same format, journals in italic and abbreviated, homogenous line spacing).

All references have been checked

Round 2

Reviewer 1 Report

- In the text, all amino acids are expressed as g/100 g of protein, except tryptophan, which is expressed in g/100 g of HPI, see lines 31 and 464, in both cases is 0.22 g/100 g of protein. On the other hand, in table 2, all the amino acids are expressed in g/100 g of protein, including tryptophan, 0.26g/100g of protein. Do you have an explanation for this difference in criteria regarding trytophan, or is it an error.

- Lines 98-99; 106; 109; 113; 247-248; 255 all the words appear together, without spaces, and it is not understood

Author Response

Response to Reviewer#1 comments R2

- In the text, all amino acids are expressed as g/100 g of protein, except tryptophan, which is expressed in g/100 g of HPI, see lines 31 and 464, in both cases is 0.22 g/100 g of protein. On the other hand, in table 2, all the amino acids are expressed in g/100 g of protein, including tryptophan, 0.26g/100g of protein. Do you have an explanation for this difference in criteria regarding trytophan, or is it an error.

Response: Thank you for your valuable comment, All amino acids expressed in g/100 g. The mistake has been corrected in the manuscript to be uniform units

- Lines 98-99; 106; 109; 113; 247-248; 255 all the words appear together, without spaces, and it is not understood

Response: The mistakes have been fixed accordingly. This mistake might be happened in pdf file because in word file everything is OK.

Reviewer 2 Report

The authors are to be commended for the improvements made in the article.  There are still some minor word usage/grammar issues.

Author Response

The authors are to be commended for the improvements made in the article.  There are still some minor word usage/grammar issues.

Response: Thank you for the a valuable comment. The all comments have been deeply considered and the manuscript has been rechecked for grammar and typewriting mistakes.
